# Perovskite-Surface-Confined Grain Growth for High-Performance Perovskite Solar Cells

**DOI:** 10.3390/nano12193352

**Published:** 2022-09-26

**Authors:** Sajid Sajid, Salem Alzahmi, Imen Ben Salem, Ihab M. Obaidat

**Affiliations:** 1Department of Chemical & Petroleum Engineering, United Arab Emirates University, Al Ain P.O. Box 15551, United Arab Emirates; 2National Water and Energy Center, United Arab Emirates University, Al Ain P.O. Box 15551, United Arab Emirates; 3College of Natural and Health Sciences, Zayed University, Abu Dhabi P.O. Box 144534, United Arab Emirates; 4Department of Physics, United Arab Emirates University, Al Ain P.O. Box 15551, United Arab Emirates

**Keywords:** post annealing, crystallinity, morphology, thin-film, perovskite solar cell

## Abstract

The conventional post-annealing (CPA) process is frequently employed and regarded a crucial step for high-quality perovskite thin-films. However, most researchers end up with unwanted characteristics because controlling the evaporation rate of perovskite precursor solvents during heat treatment is difficult. Most perovskite thin-films result in rough surfaces with pinholes and small grains with multiple boundaries, if the evaporation of precursor solvents is not controlled in a timely manner, which negatively affects the performance of perovskite solar cells (PSCs). Here, we present a surface-confined post-annealing (SCPA) approach for controlling the evaporation of perovskite precursor solvents and promoting crystallinity, homogeneity, and surface morphology of the resulting perovskites. The SCPA method not only modulates the evaporation of residual solvents, resulting in pinhole-free thin-films with large grains and fewer grain boundaries, but it also reduces recombination sites and facilitates the transport of charges in the resulting perovskite thin-films. When the method is changed from CPA to SCPA, the power conversion efficiency of PSC improves from 18.94% to 21.59%. Furthermore, as compared to their CPA-based counterparts, SCPA-based PSCs have less hysteresis and increased long-term stability. The SCPA is a potentially universal method for improving the performance and stability of PSCs by modulating the quality of perovskite thin-films.

## 1. Introduction

Perovskite solar cells (PSCs) are becoming increasingly appealing due to their wide range of applications in photovoltaics [1,2]. In comparison to other solar cells, the power conversion efficiencies (PCEs) of PSCs have rapidly risen from 3.8% [3] to over 25% [4] in a short period of time. Fabrication of high-quality perovskite thin-films is well established as a precondition for producing highly efficient PSCs [5,6,7]. Antisolvent-aided one-step spin-coating [5] and two-step sequential spin-coating [8] are the two main and commonly utilized methods for the fabrication of perovskite thin-films. Furthermore, the preparation of perovskite thin-films necessitates a post-annealing process, during which many factors must be carefully controlled; such factors include annealing temperature, annealing time, and annealing environment, all of which have a significant impact on the optoelectronic properties of the resulting perovskite thin-films [9,10,11].

It is worth noting that the ingredients, solvents, and surroundings employed to make perovskite thin-films require particular post-annealing techniques. Precursor-solvent molecules, for example, form complexes with perovskite as coordinating solvents, resulting in a perovskite-precursor solvent intermediate phase [12]. Dimethyl formamide (DMF) and dimethyl sulfoxide (DMSO) are often employed as precursor solvents to dissolve organic and inorganic salts of perovskite. DMSO can effectively coordinate with the inorganic salt of the perovskite and form a relatively stable perovskite-DMSO intermediate phase, which can retard the crystallization rate and results in perovskite thin-films with large grains and smooth surface morphology [13]. The constituents of the perovskite, on the other hand, are difficult to dissolve entirely in just DMSO. Because the organic and inorganic salts of perovskite have different solubilities, and DMF is more volatile than DMSO, the perovskite precursor containing only DMF solvent has previously been shown to result in a rapid but uneven nucleation and crystallization rate, resulting in incomplete surface coverage and rough morphology in perovskite thin-films [13,14]. Therefore, mixtures of co-solvents are mostly employed to produce perovskite precursor solutions for thin-film depositions [15,16,17,18,19,20]. Furthermore, the precursor solvents evaporate along the grain boundaries of the perovskite during the annealing process, and the mixture of perovskite and precursor solvents transforms into perovskite and solvent vapors [21,22,23,24,25,26,27,28,29,30]. In this context, the rate at which molecules from the perovskite precursor evaporate is critical for modulating the quality of the resultant perovskite thin-films.

A simple SCPA technique was used in this work to control the rate of evaporation of the precursor solvents in the perovskite during the post-annealing process. When the SCPA method was used to make perovskite thin-films, the evaporation rate of the precursor solvents was reduced because it is difficult for solvent molecules to evaporate directly from the confined perovskite’s surface. The volatilization of the precursor solvents was reduced as a result, and high-quality perovskite layers were fabricated in a controlled manner. In the CPA approach, on the other hand, the leftover precursor solvents evaporated immediately, and the perovskite solid phase formed quickly, resulting in rapid nucleation with high nucleation density, and the perovskite thin-films did not have enough time for favorable grain formation. The PSCs prepared using the SCPA technique had high PCEs of 21.59%, compared to 18.94% for CPA-based devices. In addition to having higher PCEs, SCPA-based PSCs had better stability and less hysteresis than CPA-based PSCs.

## 2. Results and Discussions

To examine perovskite crystal formation and thin-film morphology, perovskite thin-films were prepared using CPA and SCPA, as shown in Figure 1. Because of the open surface of the perovskite in the CPA, solvents evaporate quickly, resulting in undesired thin-film morphologies. Thin-films containing perovskite solvents in the SCPA, on the other hand, were covered with a Teflon sheet to control the evaporation rate of the precursor solvents and develop desirable perovskite thin-films. In addition, the post-annealing time and temperature for perovskite thin-films must be taken into account. The post-annealing time and temperature for the CPA were kept at 15 min and 130 °C, respectively, as reported in the literature for high-quality perovskite layers [5,15,16,19,20,31]. SCPA-1, SCPA-2, and SCPA-3, on the other hand, had their post-annealing times adjusted to 15 min, 25 min, and 35 min, respectively, while maintaining a constant temperature of 130 °C.

Scanning electron microscopy (SEM) was used to examine the morphologies of the thin-films obtained with CPA and SCPA. Figure 2 shows that perovskite thin-films made with the SCPA technique, specifically SCPA-2, not only have full surface coverage and homogeneity, but also large grains with fewer boundaries. Appendix A shows the average grain size for the as-prepared perovskite thin-films, which were calculated to be 256 nm, 317 nm, 510 nm, and 635 nm for the CPA, SCPA-1, SCPA-3, and SCPA-2, respectively. The low-resolution SEM images in Appendix A are useful for seeing the broad area covered by perovskite grains. As can be seen, the surface of the CPA-grown perovskite thin-films is made up of tiny grains with numerous grain boundaries, which may trap photogenerated charge-carriers. The SCPA-2 method, on the other hand, produces a smooth and compact perovskite layer. The smaller nanoparticles vanish, grain sizes grow, and a pinhole-free perovskite layer covers the surface of the SnO_2_/FTO substrate.

As shown in Figure 3, the surface morphologies of the perovskite layers produced with CPA and SCPA were further studied using atomic force microscopy (AFM). The root-mean-square (rms) value for the SCPA-2 is lowered from 11.61 nm to 4.31 nm, showing that the as-fabricated perovskite thin-films have a very smooth surface. Because the density of the grain boundary is decreased, which is good for efficient charge transportation, a high photovoltaic performance of the PSCs is expected. X-ray diffraction (XRD) measurements were used to further explore the crystallinities of the perovskite thin-films. Perovskite thin-films produced with CPA and SCPA had the typical perovskite peaks at 14.1°, 28.4°, 31.8°, 40.5°, and 43.1°, which correspond to crystal planes (001), (002), (012), (022), and (003), respectively, as shown in Appendix A. The high-intensity peaks, however, indicate that the perovskite thin-films are highly crystalline in the case of the SCPA approach. In addition, the absence of residual PbI_2_ in the SCPA process shows that the perovskite precursor is totally transformed into the desired perovskite structure (here MA_0.85_FA_0.15_PbI_3_). The undesirable peak at 12.7° in the perovskite crystals prepared with the CPA method is displayed in Appendix A. This suggests that the SCPA approach is beneficial to control the evaporation rate of the precursor solvents and the volatile organic cations. From the XRD measurements, the tendency of the improved crystallinity of the perovskite thin-films obtained with SCPA, in comparison to the CPA one, are consistent with the results obtained from their SEM and AFM images. Figure 1 shows the perovskite grain formation during the CPA and SCPA methods. In fact, the precursor solvent molecules intercalate with perovskite as coordinating solvents, resulting in a perovskite-precursor solvent intermediate phase. The precursor solvents emerged along the grain boundaries of the perovskite during the annealing procedure, and the mixture of perovskite and precursor solvents was transformed into perovskite and solvent vapors.

UV–visible (UV–vis) absorption, steady state photoluminescence (PL), and time-resolved PL spectroscopies were further used to explore the light-harvesting capabilities and charge-carrier dynamics inside perovskite thin-films prepared on glass substrates. SCPA-2 has higher absorption in the region of 416.5 nm to 525 nm than other methods, as seen in Figure 4a. The highly crystalline thin-films with large grains and minimal grain boundaries are credited with better light harvesting. When the substrate surface is coated with small grains that have some gaps at the boundaries, as is the case with CPA methods, we believe that some light just flows through the void regions where there is no perovskite. When using the SCPA approach, on the other hand, perovskite thin-films are produced that have big grains and few boundaries and absorb most of the light [32]. Furthermore, the increased absorption of the SCPA-2-based perovskite layer compared to the CPA suggests fewer defect states inside the bandgap, which is advantageous for harvesting more incident light and suggesting of excellent photovoltaic performance.

The steady-state PL of the as-prepared perovskite thin-films is shown in Figure 4b. When compared to the other thin-films, the perovskite thin-films prepared using the SCPA-2 approach have the strongest peak intensity, suggesting fewer defect states. This can be explained by looking at the main photogenerated charge processes (Figure 4c). Charge-carriers may not be trapped within grain boundaries/defect states following photoexcitation, resulting in reduced non-radiative recombination. At appropriate charge collecting layers, free charge-carriers can be collected. The CPA-based perovskite thin-films contain smaller grains with many boundaries, which trigger photogenerated charge-carrier recombination, resulting in high radiative recombination and the lowest PL peaks. SCPA-based perovskite thin-films, on the other hand, have compact, smooth, and big grains with few boundaries, resulting in low radiative recombination, and so have high PL peaks. As illustrated in Figure 4d, the charge-carrier recombination mechanism was further investigated by measuring the TRPL of the prepared samples. CPA, SCPA-1, SCPA-2, and SCPA-3 had bulk lifetimes (τ_2_) of 164.91 ns, 278.06 ns, 415.88 ns, and 338.62 ns, respectively, according to TRPL measurements. This suggests that perovskite thin-films made using the SCPA method have a longer lifespan for easily collected photogenerated charge-carriers.

Two types of planar PSCs were produced utilizing the CPA and SCPA-2 techniques, as shown in Appendix A, to investigate the effect of as-prepared perovskite thin-films on device photovoltaic performance. Figure 5 shows cross-sectional SEM images and current–voltage (J–V) characteristic curves of the as-prepared PSCs. The grains of perovskite thin-film in the cross-sectional SEM images of the as-prepared PSCs match the results obtained in the surface morphologies of Figure 2, Figure 3 and Appendix A.

The PCE for the CPA-based PSC was 18.94%, while the PCE for the SCPA-2-based PSC was 21.59% under reverse-biased conditions, according to the J–V characteristic curves. As indicated in Table 1, the SCPA-2-based devices’ short-circuit current density (J_sc_), open-circuit voltage (V_oc_), fill factor (FF), and PCE are all better than the PSCs obtained using the CPA technique. The J_sc_ rises from 23.26 mA cm^−2^ to 24.81 mA cm^−2^, V_oc_ from 1.096 V to 1.113 V, and the FF rises from 74.23% to 78.18%. The high performance of the PSCs prepared with the SCPA-2 approach can be attributed to the large grains with fewer grain boundaries, which lead to improved photon harvesting and reduced charge-carrier recombination [33]. In addition, the J–V characteristic curves of as-prepared PSCs were traced under forward and reverse-bias conditions to investigate hysteresis phenomena. As shown in Figure 5c,d, all PSCs exhibit hysteresis; however, the SCPA-2-based device shows less hysteresis than the CPA-based PSC. The SCPA-2-based PSC’s mild hysteresis effect is ascribed to less charge-carrier trapping at grain boundaries. Because SCPA-2-based perovskite thin-films have fewer grain boundaries and smoother surface morphologies than other perovskite thin-films, charge-carriers will pass through these crystals more effectively, resulting in a lower hysteresis effect. Small grains with multiple boundaries in CPA-based perovskite thin-films, on the other hand, will trap charge-carriers, and result in a significant hysteresis effect in the device.

External quantum efficiencies (EQEs) of the as-prepared devices are shown in Figure 6a. SCPA-2 and CPA-based devices had integrated J_sc_ values of 24.18 mA.cm^−2^ and 22.99 mA cm^−2^, respectively. The integrated J_sc_ values are nearly identical to those derived from J–V curves, implying that the EQEs and J–V measurements are consistent. Electrical impedance spectroscopy (EIS) was used to investigate the charge-carrier dynamics within PSCs under photoexcitation with a reverse bias of 0.9 V and a frequency range of 1 Hz to 1 MHz. Figure 6b shows Nyquist plots with equivalent circuits to evaluate the recombination resistances (R_r_) of the charge-carriers. The charge-transporting layers for the as-prepared PSCs were kept the same in this study, therefore the R_r_ was only affected by the perovskite thin-film. Figure 6b shows that the SCPA-2 based PSC has a higher R_r_ of 108.26 KΩ when compared to the R_r_ of the CPA-based device (97.61 KΩ). The high R_r_ of the SCPA-2 based device efficiently resists charge-carrier recombination, which is in line with the previous findings.

In addition to high PCE, long-term stability is another desirable characteristic of the PSCs. As shown in Figure 6c, the steady-state J_sc_ and PCE of the SCPA-2-based PSCs were first tracked at maximum power at a bias voltage of 0.95 V at room temperature under AM 1.5 G. The SCPA-2-based device displays a stabilized PCE of 21.36% after 236 s of continuous illumination, which is very close to the greatest PCE of 21.59% for the same PSC. Furthermore, under AM 1.5 G irradiation, the stabilities of unsealed-PSCs prepared using the CPA and SCPA-2 methods were assessed every 72 h for 720 h. The measured devices were maintained in a glass oven at 65% humidity and room temperature after each stability test. Figure 6d shows how the SCPA-2-based PSCs outperformed the CPA-based devices in terms of stability. The efficiency of SCPA-2-based PSCs reduced slightly from 21.54% to 21.01% after 720 h of testing, preserving 97% of the original PCE. Under the same testing conditions, the PCE of CPA-based PSCs dropped dramatically from 18.79% to 4.74%, showing poor stability and rapid degradation. Because CPA-based perovskite thin-films have a rough surface morphology and tiny grains with multiple boundaries, the degradation of CPA-based PSCs is linked to fast erosion. Meanwhile, as shown in Appendix A, the statistical data of PSCs based on CPA and SCPA-2 (10 devices for each) were analyzed. The statistical data shows that PSCs prepared with SCPA-2 have higher photovoltaic metrics and better repeatability than PSCs prepared using CPA. For PSCs prepared using the SCPA-2 technique, all photovoltaic metrics are within a narrow range, whereas CPA-based devices have widely dispersed parameters. With an average J_sc_ of 22.58 mA.cm^−2^, V_oc_ of 1.03 V, and FF of 73.62%, the PCEs for 10 CPA-based PSCs varied widely from 15.87% to 18.94%. In contrast, the PCEs for the PSCs with SCPA-2 were distributed in a narrow range between 20.14% and 21.59%. With an average J_sc_ of 24.21 mA.cm^−2^, V_oc_ of 1.102 V, and FF of 77.72%, the average PCE reaching 20.73%. The above results demonstrate that the SCPA-2 technique is a simple method for producing high-quality perovskite layers and, as a result, high photovoltaic performance in PSCs.

## 3. Conclusions

A facile and effective SCPA strategy is used for controlling the evaporation of perovskite precursor solvents and promoting crystallinity, homogeneity, and surface morphology of the resulting perovskite thin-films. The SCPA method not only modulates the evaporation of residual precursor solvents, resulting in pinhole-free perovskite layers with large grains and fewer grain boundaries, but it also reduces recombination sites and facilitates the transport of photogenerated charges in the resulting perovskite thin-films. The SCPA-based devices delivered PCEs of up to 21.59%, higher than those of the control devices (18.94%). Furthermore, the SCPA-based PSCs showed less hysteresis and more long-term stability compared to the control devices.

## 4. Materials and Methods

### 4.1. Preparation of the Precursor Solutions

A 1 mL colloidal solution of SnO_2_ was diluted with 5 mL of deionized water to make the precursor solution for SnO_2_. For the preparation of the perovskite precursor, which was stirred for 2 h in the glove box, 0.85 M of methylammonium iodide (MAI), 0.15 M of formamidinium iodide (FAI), and 1.025 M of lead iodide (PbI_2_) were dissolved in 600 mg/78 mg of anhydrous DMF/DMSO. The solution for the hole-transporting material was made up of 80 mg of Spiro-OMeTAD in 1 mL of chlorobenzene, 28.5 μL of tert-butylpyridine in 1 mL of chlorobenzene, and 8.75 mg mL^−1^ of lithium bis-(trifluoromethanesulfonyl)-imide. In the glove box, this mixture was stirred for 6 h.

### 4.2. Fabrication of the Perovskite Solar Cells

The FTO-substrates were cleaned in sequence by sonicating them for 20 min each in detergent solution, deionized water, acetone, and isopropyl alcohol. The FTO-substrates were treated with UV–ozone for 15 min following nitrogen blow-drying. On the well cleaned FTO-substrate, the previously prepared SnO_2_ solution was dropped and spun at 3000 rpm for 15 s. For 30 min, the SnO_2_-coated FTO-substrates were heated at 150 °C. After cooling to room temperature, the FTO/SnO_2_ substrates were then transferred to a glove box. On FTO/SnO_2_ substrate, the perovskite solution was spin-coated at 4000 rpm for 30 s. During the final 15 s of spinning, 0.8 mL of diethyl ether was dropped onto the substrates. After that, two techniques—CPA and SCPA—were employed for post-annealing in order to assess the quality of the perovskite thin-films. Unlike the SCPA approach, where the surface of the perovskite was covered with a Teflon sheet, the CPA approach left the surface of the perovskite material uncovered. The CPA was post-annealed for 15 min at a temperature of 130 °C. While keeping the temperature at 130 °C, the post-annealing times for the SCPA were changed to 15, 25 and 35 min, respectively. Both CPA and SCPA-based perovskite films were spin-coated with Spiro-OMeTAD solution at 4000 rpm for 30 s after cooling to room temperature. The solar cells were then finished by evaporating 80 nm gold electrodes onto the as-prepared thin-films. For a 0.1 cm^2^ active area, a metal mask was employed in each device.

### 4.3. Device Characterizations

Morphological investigation was done with a scanning electron microscope (Hitachi S-4800, Tokyo, Japan) and an atomic force microscope (Aglient Keysight AFM-5500, Santa Clara, CA, USA). Using an X-ray diffractometer (Bruker D8 Advance, Cu-Kα radiation of λ = 0.15406 nm), the crystallinity of the perovskite was investigated. The absorption spectrum was measured using a UV–Vis spectrophotometer (UV-2600). Using Edinburg PLS 980, the steady PL spectra of the produced perovskite films were examined. The TRPL decay of the perovskite films was measured using a transient-state spectrophotometer (Edinburg Ins. F900, Edinburg, UK) under a 485 nm laser. Under AM 1.5 G illumination with a power intensity of 100 mW cm^−2^, the J–V characteristic curves were measured with a source meter (Keithley 2400, Cleveland, OH, USA) using forward (−0.1 to 1.2 V) or reverse (1.2 to −0.1 V) scans from a solar simulator (XES-301S + EL-100). The delay duration was set to 10 ms, and the step voltage was set to 12 mV. The EQE was calculated using the QE-R system (Enli Tech., Atlanta GA, USA). The EIS measurement was carried out using an electrochemical workstation (Zahner Zennium, Kansas City, MI, USA).

## Figures and Tables

**Figure 1 nanomaterials-12-03352-f001:**
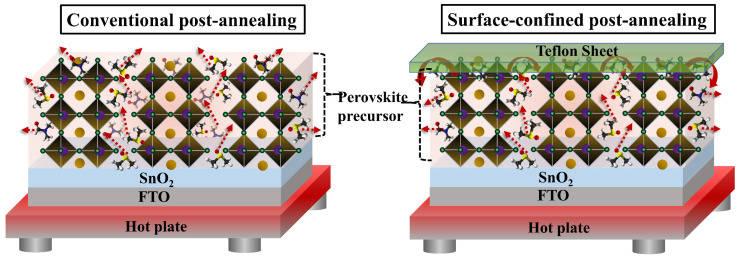
Schematic illustration of conventional post-annealing (CPA) and surface-confined post-annealing (SCPA) proposed for the preparation of perovskite thin-films.

**Figure 2 nanomaterials-12-03352-f002:**
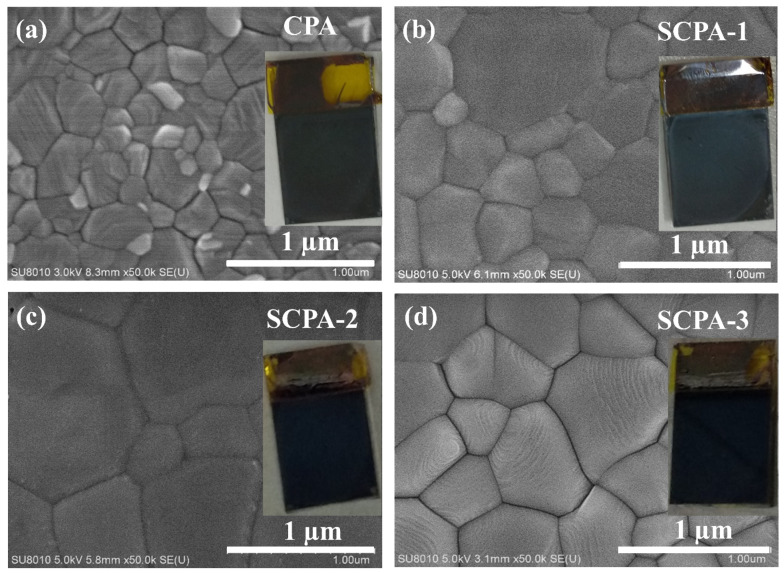
SEM images of the as-prepared perovskite thin-films with CPA (**a**) and SCPA (**b**–**d**). Insets are the digital photos of each perovskite/SnO_2_/FTO-substrate.

**Figure 3 nanomaterials-12-03352-f003:**
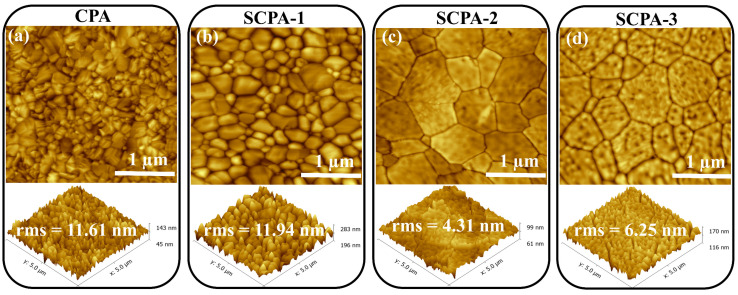
AFM images with rms values of the as-prepared perovskite thin-films with CPA (**a**) and SCPA (**b**–**d**).

**Figure 4 nanomaterials-12-03352-f004:**
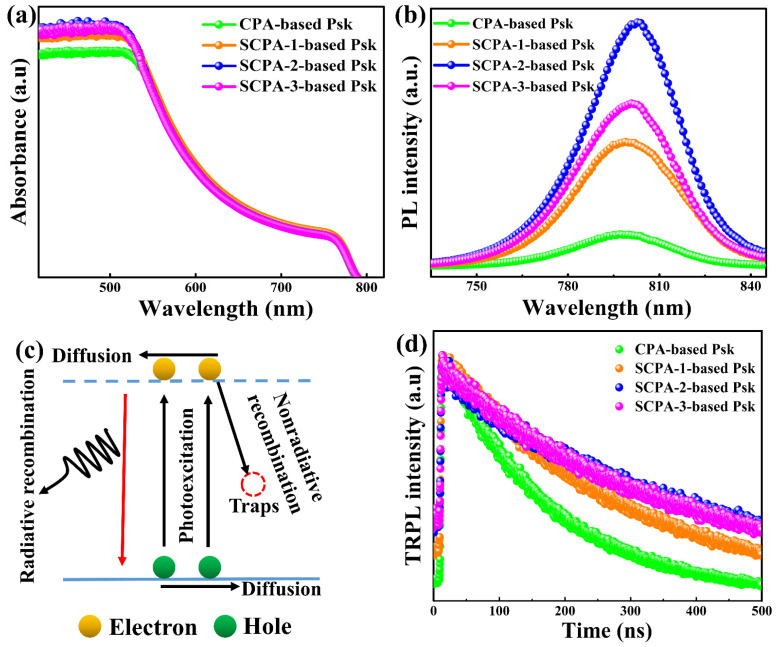
The optical and electronic properties of perovskite thin-films made using CPA and SCPA-2. PL-spectra (**a**), UV-Vis spectra (**b**), Charge-carrier dynamics in perovskites (**c**), and TRPL-spectra (**d**).

**Figure 5 nanomaterials-12-03352-f005:**
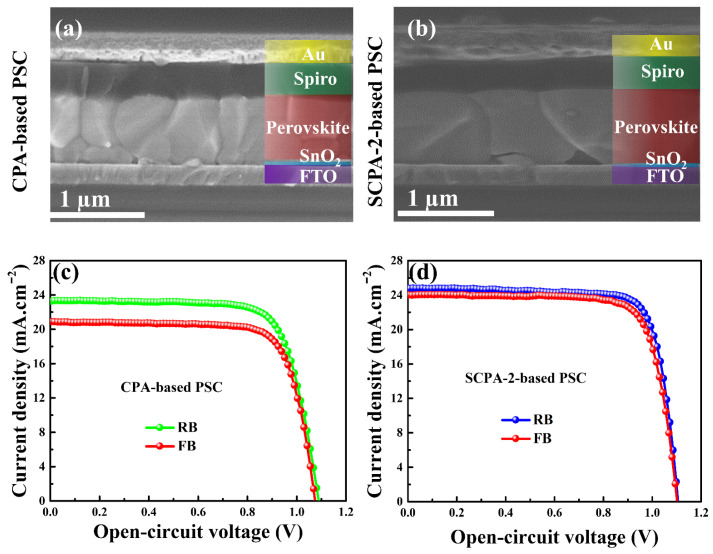
Cross-sectional SEM images of the PSCs prepared with CPA (**a**) and SCPA-2 (**b**). Current–voltage characteristic curves of PSCs prepared with CPA (**c**) and SCPA-2 (**d**).

**Figure 6 nanomaterials-12-03352-f006:**
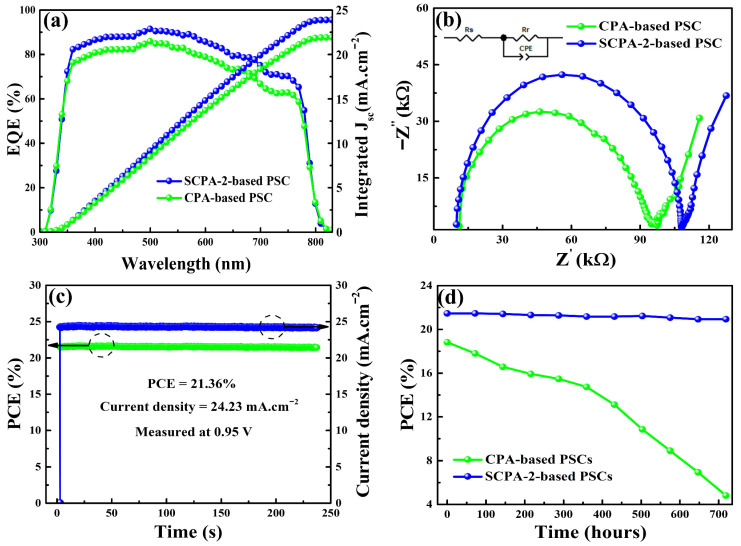
EQEs with integrated Jsc (**a**), EIS (**b**), and stability analyses (**c**,**d**) of the PSCs prepared using the CPA and SCPA-2 method.

**Table 1 nanomaterials-12-03352-t001:** Photovoltaic parameters of the as-prepared PSCs under forward bias (FB) and reverse bias (RB).

PSCs Prepared with	Scan Condition	V_oc_ (V)	J_sc_ (mA.cm^−2^)	FF (%)	PCE (%)
CPA	RB	1.096	23.26	74.23	18.94
	FB	1.070	20.88	76.28	17.04
SCPA-2	RB	1.113	24.81	78.18	21.59
	FB	1.109	24.02	76.78	20.45

## Data Availability

All the data presented in the manuscript can be obtained from the corresponding authors by reasonable request.

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
