# Peer review of "Perovskite-Surface-Confined Grain Growth for High-Performance Perovskite Solar Cells"

_nanomaterials, 2022, doi:10.3390/nano12193352_

Round 1

Reviewer 1 Report

The manuscript „Perovskite-surface-confined grain growth for high-performance 2 perovskite solar cells“ by Sajid Sajid et al presents a new of post-annealing, which results in better quality of perovskite thin-film and improvement of photovoltaic characteristics. The article may be interesting for researchers working on development of perovskite solar cells. However, the manuscript contains some inaccuracies, which must be corrected.

Page 2 rows 70-71: … resulting in rapid nucleation with high density …

It is not clear about what density the authors are talking. Is it density of nucleation centers?

Reviewer’s comment: The part of first paragraph of Results and Discussion section (page 2, rows 71-92) presents nearly the same statements on post-annealing process like those already presented in Introduction section, so they could be removed or presented in the Introduction section. The text in rows 71-92 does not present results of current work.

P.3 row 114: Insets are the digital images of each perovskite/SnO2/FTO-substrate

Reviewer’s comment: It would be better to write “digital photos” instead of “digital images”

P.3 row 121: Perovskite thin-films produced with CPA and SCPA had the typical perovskite miller indices

Reviewer’s comment: Thin-films don’t have Miller indices. An XRD pattern of the film can show peaks, which can be attributed to perovskite peaks with following indices.

P.4 rows 124-126: The absence of residual PbI2 in the SCPA process, however, indicates  that the perovskite thin-films are highly crystalline …

Reviewer’s comment: The absence of residual PbI2 has no relation with crystallinity. Sharp peaks attributable to perovskite phase and absence of very wide peaks characteristic of amorphous material confirm high crystallinity of perovskite film.

Reviewer’s comment: The text on page 4 rows 142-150 almost fully reproduces what has already been written about on the page 2.

Reviewer’s comment: Measurements errors for JSC, VOC, FF and PCE are not presented while differences between those values for samples prepared by CPA and SCPA-2 method are not so much significant.

P.9 row 300: Bruker D8 Advace, Cu-K radiation of 0.15406 nm

Reviewer’s comment: Did authors have used monochromator on primary x-ray? Cu Kalpha1 radiation has wavelength of 0.15406 nm, however if the primary x-ray beam was not monochrameted the radiation Cu Kalpha 1,2 was used and the wavelength of the latter radiation is 0.154184 nm.

P.10

16. H. Huang, H. Yan, M. Duan, J. Ji, X. Liu, H. Jiang, B. Liu, S. Sajid, P. Cui, Y. Li, M. Li, TiO2 surface

17. A.M. Elseman, A.E. Shalan, S. Sajid, M.M. Rashad, A.M. Hassan, M. Li, Copper-Substituted Lead Perovskite Materials Con-359 structed with Different Halides for Working (CH3NH3)2CuX4-Based Perovskite Solar Cells from Experimental and Theoretical 360 View, ACS Appl. Mater. Interfaces. (2018). https://doi.org/10.1021/acsami.8b00495.

20. H. Huang, X. Liu, M. Duan, J. Ji, H. Jiang, B. Liu, S. Sajid, P. Cui, D. Wei, Y. Li, M. Li, Dual Function of Surface Alkali-Gas 368 Erosion on SnO2 for Efficient and Stable Perovskite Solar Cells, ACS Appl. Energy Mater. 3 (2020) 5039–5049. 369 https://doi.org/10.1021/acsaem.0c00563.

Reviewer’s comment: Subscripts must be used in the chemical formula.

The authors repeat many times that the morphology of thin layers obtained by the SCPA-2 method improves the morphology, the grains increase, and the density of the boundaries decreases.

Perhaps it would be better to present the results obtained and, at the end of the chapter, to explain the improvement in optical and photoelectric properties by the positive changes in the morphology, the grains size and the density of the grains boundaries in the samples formed by the SCPA-2 method. This would avoid constant repetition, so that the positive changes in the functional properties of the perovskite layer, formed by the SCPA-2 method, are the result of an increase in the grains size and a decrease in the density of the grain boundaries.

Author Response

Dear Ms. Vanessa Zhang,

We are very thankful for your kind letter and decision on the Manuscript “Perovskite-surface-confined grain growth for high-performance perovskite solar cells". According to the reviewer’s comments and suggestions, we made the necessary changes to the manuscript. After modifying the manuscript, we hope that your prestigious journal will accept our manuscript.

Comments and Suggestions for Authors

The manuscript “Perovskite-surface-confined grain growth for high-performance perovskite solar cells“ by Sajid Sajid et al presents a new of post-annealing, which results in better quality of perovskite thin-film and improvement of photovoltaic characteristics. The article may be interesting for researchers working on development of perovskite solar cells. However, the manuscript contains some inaccuracies, which must be corrected.

  1. Page 2 rows 70-71: … resulting in rapid nucleation with high density …

It is not clear about what density the authors are talking. Is it density of nucleation centers?

Reply: Thank you very much for highlighting this point. Actually this is “high nucleation density”, as mentioned in the revised manuscript page 2, line 71.

  1. Reviewer’s comment: The part of first paragraph of Results and Discussion section (page 2, rows 71-92) presents nearly the same statements on post-annealing process like those already presented in Introduction section, so they could be removed or presented in the Introduction section. The text in rows 71-92 does not present results of current work.

Reply: Thank you for your valuable suggestion. The paragraph of the Results and Discussion is modified accordingly. The mentioned text in rows 71-75 is about the power conversion efficiencies, stability and hysteresis of the as-prepared PSCs, which are supported by the experimental results. The following rows are modified i.e. rows 86-92 represent the methods used for the fabrication of the perovskite layers.

  1. 3 row 114: Insets are the digital images of each perovskite/SnO2/FTO-substrate

Reviewer’s comment: It would be better to write “digital photos” instead of “digital images”

Reply: We made the changes accordingly, as can be seen in the revised manuscript page 3, line 115.

  1. 3 row 121: Perovskite thin-films produced with CPA and SCPA had the typical perovskite miller indices

Reviewer’s comment: Thin-films don’t have Miller indices. An XRD pattern of the film can show peaks, which can be attributed to perovskite peaks with following indices.

Reply: We agreed with your comment. We rearranged the statements as “CPA and SCPA had the typical perovskite peaks at 14.10, 28.40, 31.80, 40.50, and 43.10, which correspond to crystal planes (001), (002), (012), (022), and (003), respectively……..”, as can be seen on page 3 and 4 of revised manuscript.

  1. 4 rows 124-126: The absence of residual PbI2 in the SCPA process, however, indicates that the perovskite thin-films are highly crystalline …

Reviewer’s comment: The absence of residual PbI2 has no relation with crystallinity. Sharp peaks attributable to perovskite phase and absence of very wide peaks characteristic of amorphous material confirm high crystallinity of perovskite film.

Reply: Yes you are right. In fact, from the absence of PbI2, we mean the complete conversion of perovskite precursor to the desirable perovskite thin-film. This statement is rearranged in the revised manuscript, as can be seen on page 4.

  1. Reviewer’s comment: The text on page 4 rows 142-150 almost fully reproduces what has already been written about on the page 2.

Reply: Thank you for this point. We agreed with your comments, therefore, we removed the mentioned rows in order to avoid repetition.

  1. Reviewer’s comment: Measurements errors for JSC, VOC, FF and PCE are not presented while differences between those values for samples prepared by CPA and SCPA-2 method are not so much significant.

Reply: The statistical data of the as-prepared devices in figure S5, demonstrate that CPA-based devices have widely dispersed parameters. With an average Jsc of 22.58 mA.cm-2, Voc of 1.03 V, and FF of 73.62%, the PCEs for 10 CPA-based PSCs distributed widely from 15.87% to 18.94%. In contrast, the PCEs for the PSCs with SCPA-2 are distributed in a narrow range between 20.14% and 21.59%. In addition, we provided table S (supplementary file) containing all photovoltaic parameters, which clearly shows the significant differences between the photovoltaic parameters.

  1. 9 row 300: Bruker D8 Advance, Cu-K radiation of 0.15406 nm

Reviewer’s comment: Did authors have used monochromator on primary x-ray? Cu Kalpha1 radiation has wavelength of 0.15406 nm, however if the primary x-ray beam was not monochrameted the radiation Cu Kalpha 1,2 was used and the wavelength of the latter radiation is 0.154184 nm.

Reply: We used X-ray diffractometer (Bruker D8 Advance, Cu-Kα radiation of λ = 0.15406 nm) with monochromator on primary x-ray.

  1. 10
  2. H. Huang, H. Yan, M. Duan, J. Ji, X. Liu, H. Jiang, B. Liu, S. Sajid, P. Cui, Y. Li, M. Li, TiO2 surface
  3. A.M. Elseman, A.E. Shalan, S. Sajid, M.M. Rashad, A.M. Hassan, M. Li, Copper-Substituted Lead Perovskite Materials Con-359 structed with Different Halides for Working (CH3NH3)2CuX4-Based Perovskite Solar Cells from Experimental and Theoretical 360 View, ACS Appl. Mater. Interfaces. (2018). https://doi.org/10.1021/acsami.8b00495.
  4. H. Huang, X. Liu, M. Duan, J. Ji, H. Jiang, B. Liu, S. Sajid, P. Cui, D. Wei, Y. Li, M. Li, Dual Function of Surface Alkali-Gas 368 Erosion on SnO2 for Efficient and Stable Perovskite Solar Cells, ACS Appl. Energy Mater. 3 (2020) 5039–5049. 369 https://doi.org/10.1021/acsaem.0c00563.

Reviewer’s comment: Subscripts must be used in the chemical formula.

Reply: All references are modified accordingly.

  1. The authors repeat many times that the morphology of thin layers obtained by the SCPA-2 method improves the morphology, the grains increase, and the density of the boundaries decreases.

Perhaps it would be better to present the results obtained and, at the end of the chapter, to explain the improvement in optical and photoelectric properties by the positive changes in the morphology, the grains size and the density of the grains boundaries in the samples formed by the SCPA-2 method. This would avoid constant repetition, so that the positive changes in the functional properties of the perovskite layer, formed by the SCPA-2 method, are the result of an increase in the grains size and a decrease in the density of the grain boundaries.

Reply: Thank you very much for your insightful suggestions. In fact, each result should be supported by a reasonable explanation, so we think the manuscript's order is appropriate. However, we made modification in order to avoid repetition of the words.

Reviewer 2 Report

The work by Sajid Sajid et al. is robust and provides new insights into the role of controlled solvent evaporation in the stabilized efficiency of planar perovskite solar cells. The comparison between surface-confined post-annealing approach and standard solvent evaporation has not been studied thoroughly and this is a very neat study showing that the first contributes positively to the performance of the solar cells. I would advise the authors, if possible, to add FT-IR data to understand the composition of the thin films in the time and to figure out if intermediate phases with solvent are forming. This could also give more insights into the mechanism. Additional, if available, report a screening upon the temperature with a fixed time would be insightful.

Author Response

Comments and Suggestions for Authors

The work by Sajid Sajid et al. is robust and provides new insights into the role of controlled solvent evaporation in the stabilized efficiency of planar perovskite solar cells. The comparison between surface-confined post-annealing approach and standard solvent evaporation has not been studied thoroughly and this is a very neat study showing that the first contributes positively to the performance of the solar cells. I would advise the authors, if possible, to add FT-IR data to understand the composition of the thin films in the time and to figure out if intermediate phases with solvent are forming. This could also give more insights into the mechanism. Additional, if available, report a screening upon the temperature with a fixed time would be insightful.

Reply: We are very thankful for your valuable suggestions. Since the main focus of our manuscript is on the development of high-quality perovskite layers using a simple physical technique, we believe that characterizations such as SEM, AFM, XRD, UV-Vis, PL, and TRPL are appropriate to demonstrate the quality of the produced thin-films through conventional post-annealing and surface-confined post-annealing. As we mentioned in the Results and Discussion on page 2 that 130  is the most suitable annealing temperature for the fabrication of the perovskite layer (https://doi.org/10.1016/j.solener.2021.08.015, https://doi.org/10.1002/adma.201707583 etc.), therefore, we kept this value unchanged. Once again, we are very thankful for your consideration.  
